# Data Governance and Sovereignty in Urban Data Spaces Based on Standardized ICT Reference Architectures

**Silke Cuno [1], Lina Bruns [1], Nikolay Tcholtchev [1,\*], Philipp Lämmel [1] and Ina Schieferdecker [1,2]**

[1]   System Quality Center (SQC), Fraunhofer Institute for Open Communication Systems (FOKUS),
     10589 Berlin, Germany; silke.cuno@fokus.fraunhofer.de (S.C.); lina.bruns@fokus.fraunhofer.de (L.B.);
     philipp.laemmel@fokus.fraunhofer.de (P.L.); ina.schieferdecker@tu-berlin.de or
     ina.schieferdecker@fokus.fraunhofer.de (I.S.)

[2]   Quality Engineering of Open Distributed Systems (QDS), Technical University of Berlin (TU Berlin),
     10587 Berlin, Germany

\*    Correspondence: nikolay.tcholtchev@fokus.fraunhofer.de; Tel.: +49-30-3463-7175

**Abstract:** European cities and communities (and beyond) require a structured overview and a set of tools as to achieve a sustainable transformation towards smarter cities/municipalities, thereby leveraging on the enormous potential of the emerging data driven economy. This paper presents the results of a recent study that was conducted with a number of German municipalities/cities. Based on the obtained and briefly presented recommendations emerging from the study, the authors propose the concept of an Urban Data Space (UDS), which facilitates an eco-system for data exchange and added value creation thereby utilizing the various types of data within a smart city/municipality. Looking at an Urban Data Space from within a German context and considering the current situation and developments in German municipalities, this paper proposes a reasonable classification of urban data that allows the relation of various data types to legal aspects, and to conduct solid considerations regarding technical implementation designs and decisions. Furthermore, the Urban Data Space is described/analyzed in detail, and relevant stakeholders are identified, as well as corresponding technical artifacts are introduced. The authors propose to setup Urban Data Spaces based on emerging standards from the area of ICT reference architectures for Smart Cities, such as DIN SPEC 91357 "Open Urban Platform" and EIP SCC. In the course of this, the paper walks the reader through the construction of a UDS based on the above-mentioned architectures and outlines all the goals, recommendations and potentials, which an Urban Data Space can reveal to a municipality/city. Finally, we aim at deriving the proposed concepts in a way that they have the potential to be part of the required set of tools towards the sustainable transformation of German and European cities in the direction of smarter urban environments, based on utilizing the hidden potential of digitalization and efficient interoperable data exchange.

**Keywords:** data governance; data sovereignty; urban data spaces; ICT reference architecture; Open Urban Platform

---

## 1. Introduction

The "data-driven transformation" influences the economy and society in an increasing manner. This development constitutes the so-called "transformation phase" towards a global "data economy". In parallel, a continuously growing amount of data is being generated, thereby building on new technological trends such as the Internet of Things, factories of the future, artificial neural networks,

big data analytics, autonomous networked systems or Smart City reference architectures. Digital data and information provide the basis for these new technologies.

The "data economy" in that sense comprises an ecosystem of different stakeholders and market participants, such as companies, infrastructure managers, public administration, research and civil society, whose cooperation ensures that data can be made accessible and usable. In this context, the market participants/stakeholders can extract and derive value from this data by implementing and operating a variety of ICT applications/services, opening a tremendous potential for improving our everyday lives, including vital aspects such as traffic management, traffic flow optimization or remote e-health services [1].

According to the European Commission [1], public and private (service) providers can benefit enormously from the emerging new data market. Municipalities are also part of this ecosystem and also have the potential to contribute and benefit. For municipalities, the first steps would be to examine, understand and define their own specific urban data, to work out and implement the necessary processes for data provisioning and data management, to build a powerful data infrastructure to support and automate these processes, and to ensure their own municipal data sovereignty.

The enormous variety of data in the municipalities offers plenty of options/potentials in many different perspectives; for example, information and insights for integrated urban development, urban environmental protection and policy-making can be gained and the local economy strengthened, for instance, with new business models and innovative data-based ideas. Often however, systematically executed overviews and studies are lacking details about the data available in municipal organizations. The usage of these data is mostly restricted to limited areas in local governments. This reinforces the partially existing silo thinking within different domains and between individual departments. In addition, German and European municipalities often lack the technical infrastructure that enables a horizontal connection between the various municipal actors and supports the integrated use of data, as well as concrete municipal business models for sustainable data exploitation. Furthermore, the municipal data economy is also hampered by the fact that a regulation regarding the utilization of created, transmitted and utilized data is missing [2], and hence the question of communal data sovereignty is not sufficiently and practically answered as to enable the utilization of large amounts of municipal urban data. German and European communities/municipalities should now take action and secure their participation in the data economy. On the way to a modern, sustainable and networked city or community, communities must be accompanied and supported. Hence, a set of tools is required that will navigate cities and communities in achieving systematical introduction and adoption of digital technologies, in order to become a data driven smart urban environment. One such tool could be the concept of an Urban Data Space that is elaborated and defined as a goal of this work in the following.

The goal and contribution of this paper is to define and specify on an abstract level the concept of an Urban Data Space, in order to tackle the imminent challenges to municipalities with respect to the emerging data driven economies across Germany, Europe and beyond. Thereby, we aim at systematically describing the various data types within a smart city/community and at providing an abstract architecture, which can be instantiated in the form of various technical components, interfaces, data formats and processes towards the efficient implementation of a smart city/community. It should be a main feature of the Urban Data Space concept that it is designed in a way as to guarantee data governance and data sovereignty for the involved partners and stakeholders, with special focus on municipalities within this paper. Furthermore, the concept of an Urban Data Space should be able to accelerate the adoption and introduction of smart city technology across Europe and beyond by providing an abstract blue print and facilitating the development, replication and exchangeability of components, services, applications and process according to open standards.

The basis for the creation of the Urban Data Space, which can be understood as a "precursor" for "smart cities/communities", is a clear overview of the existing urban data, as well as an easy retrieval and integrated access to the existing urban data. The Urban Data Space should offer a comprehensive

range of municipal data as well as an overarching utilization roadmap for the data in the overall communal context.

In order to review the current situation of German municipalities, the conducted study provides an inventory analysis of municipal data and legal framework conditions in selected German municipalities. The following questions are of central importance: (1) What characterizes the urban data? (2) Which data is already available in the municipal databases of the examined model regions? (3) Which urban actors are interested in data exchange? (4) How are the legal framework conditions? (5) How can an IT/ICT architecture be designed sustainably for urban areas? Based on these questions and taking into account the German "Smart City Charter" [3] and the "Sustainable Development Goals of the 2030 Agenda" [4], the current paper formulates recommendations and guidelines that should orientate municipalities on how to efficiently design a future-proof Urban Data Space.

To summarize our approach: In order to achieve the definition of the Urban Data Space, we proceed with the following steps. First, an extensive literature review and urban data classification is provided which leads to setting the conceptual framework for the upcoming design approach. Secondly, we present a summary of the analysis of the situation in three German cities. The cities are selected based on recommendations from key stakeholder organizations in the German municipal IT and data landscape, as well as on the different size of the cities and level of their digitalization. The analysis was conducted based on structured interviews with key persons in the belonging municipality leading to the validation of a couple of hypotheses, as well as to the extraction of key factual statements and concrete recommendations/requirements. Based on these recommendations and the conceptual framework, we systematically derive the concept of an Urban Data Space thereby highlighting its benefits for cities and communities and defining a possible implementation of data governance and data sovereignty procedures.

The rest of this paper is organized as follows: Section 2 analyzes the frame (including data classification) for Urban Data Spaces and defines the proposed concept. Section 3 presents briefly the results from the study relating to UDS and conducted in multiple German cities and municipalities. Section 4 proposes an abstract design for Urban Data Spaces and analyzes the benefits for cities/municipalities, thereby clearly outlining the required steps towards a successful large-scale implementation. The following Section 5 shows how the important aspects of data governance and data sovereignty would be addressed, whilst the final Section 6 concludes this paper and presents potential future research and development directions.

## 2. Analyzing the Frame for an Urban Data Space

In the current chapter, we set the frame for understanding and specifying the notion of an Urban Data Space. A number of key aspects are discussed starting with a classification of urban data, proceeding with introducing the idea of data sovereignty as well as with a first understanding of an Urban Data Space. Furthermore, the chapter handles the identification of key stakeholders in the scope of an UDS and introduces the extremely important concept of an urban ICT reference architecture, which is at the heart of the proposed approach.

In addition to the data-owner-related classification presented below, there are various data collection/gathering/acquisition options that are currently used in the urban environment. The variety of methods used is very wide and includes statistical procedures, as well as data collection methods within the public administration (for example, by reporting obligations), but also sensor-based approaches in the context of the Internet of Things, which are increasingly used in the course of the digital urban transformation. Selected relevant examples of data collection methods in the Urban Data Space are presented in appendix A.5 in [5]. The examples are chosen to reflect the latest state of research in EU projects and national projects. It should be emphasized that besides these examples, of course, there are also the traditional methods that are currently practiced in municipalities. Good examples here are the established statistical offices and geo-data offices, whose proven data collection methods provide a rich source of data for the Urban Data Space. After this brief discussion on possible data

collection/gathering methods, the presentation on providing data for the Urban Data Space continues with further details in the coming section.

*2.1. Data Classification*

The term "urban data" refers to all types of data that are important in the urban context, regardless of the specific data origin, data management, the associated intellectual property rights and licensing requirements. Urban data may include data that extends beyond the direct local context, for example, when needed for a municipal process based on data of supra-regional or global relevance, or simply if it has general effects on the urban space/environment—for example, climate data or financial data.

The Urban Data Space refers to the entire set of data that has relevance in the urban context (economic, urban, geographical, technical, climatic, health, etc.) and is needed, generated or collected within municipal processes. The "smart city/community" concept intends to open up this data such that the municipality or the municipal companies can facilitate and accelerate the corresponding provisioning processes. The ICT-based services and applications of the municipality should also utilize this data. Data can be provided directly as a good or used as a basis for innovative services.

The proposed concept within "Data for London: A City Data Strategy" [6] treats the term "city data" as a central element of an embodiment of the Greater London as intelligent, urban, ICT-based ecosystem. Great emphasis is placed on the successful implementation of relevant open-data strategies[1]. The so-called "Data for London Strategy" plans to extend these approaches by additional types of data and corresponding data providers. This complementary data is expected to come not only from the public administration, but also from the private sector. Data providers can be urban utilities, as well as various infrastructure operators, distributors, start-ups and many more. All these data sources/providers would require a unified view and a sophisticated data exchange infrastructure, which is one of the set goals of the emerging Urban Data Space concept.

The challenges in data acquisition and provisioning in an urban environment emerge on multiple levels. Firstly, the data sources should be secured and instrumented/convinced as to make their data available to a wider range of stakeholders. This phase might include the interaction with municipal departments, municipal companies, local service providers and enterprises, but also the installation of additional/new sensors across the urban area in question, and the deployment of services for data handling and analysis. Secondly, the quality of the data should be ensured as to enable a data driven smart urban environment. Data quality includes mainly the correctness, the timeliness and the machine readability of the urban data. Thereby, the data quality might differ in its criteria depending on the type of data as classified further in this section, e.g., sensor data should be delivered within milliseconds or seconds, as to be useful for near real-time applications, whilst municipal public administration data might be adequate if delivered once in a quarter. Furthermore, the quality of the data delivery infrastructure[2] is of serious concern with respect to conformance to standards, interoperability between components, performance, scalability, stability of the hardware/software implementations, network bandwidth, cyber-security (in terms of confidentiality, integrity, and availability), data protection and data privacy given its intrinsic nature as a critical infrastructure. Moreover, the utilization of open standardized interfaces is of paramount importance as to enable future business models and avoid vendor lock-in with respect to data provisioning and utilization. In addition, an Urban Data Space should provide marketplace like means for data exchange including data trading as to stimulate the provisioning and utilization of data for future use cases and business models.

Having discussed on the overall picture and challenges regarding urban data, the following paragraphs present the specified and adopted classification of data.

---

[1] For example, data.gov.uk and data.london.gov.uk.
[2] The aspect of the quality of the data delivery infrastructure is beyond the scope of this paper and will be addressed in our future work.

### 2.1.1. Official Institutional Data

Official data refers to all data available from/for public-law institutions performing administrative tasks. Examples of such data can be found in official statistics, i.e., statistics compiled by an official institution (in particular a statistical office), or for example, through official surveying conducted by the responsible institutions. Further examples for official data in urban environments are also given by data from public offices, cadastres or municipal utility data, such as water supply data and energy data, if organized under public law.

### 2.1.2. Enterprise Data

As enterprise data, we refer to all data arising within a company. Enterprise data can be obtained within a company itself or from external sources, such as market and customer data, consumer behavior or business relationships. For example, the data from the purchase of raw materials, consumables and supplies can be commercially available for production plants and with respect to final products. Even companies can provide data as open data, as exemplified by the Open Data Portal of the Berlin energy provider (see: netzdaten-berlin.de).

A major obstacle for the exchange of enterprise data is given by the risk that the provided data might contain corporate secrets. Relevant data (in the domain of data-driven companies) could potentially include source code, algorithms or entire repositories, theoretical models, system architectures or other modelling artifacts, e.g., UML diagrams, use cases and other functional descriptions. As long as industrial property rights do not address data aspects and technical infrastructure, it is up to the company to determine to what extent certain internal enterprise data artifacts should remain protected [7].

### 2.1.3. Research Data

According to a definition of the alliance of German science organizations, research data is data "*that arises in the course of scientific projects, for example based on digitalization, desktop research, experiments, measurements, surveys or questionnaires*" [8]. Research data can include measurement data, laboratory results, audiovisual information, data from studies, samples and probes that originate from, are developed or evaluated in the course of scientific work, as well as methodological test procedures, such as questionnaires, software or simulations [9]. In the scope of providing research data into a research data space, various German science organizations and the "Council for Information Infrastructures" (RfII) are currently working on setting up a German-wide "National Research Data Infrastructure" (NFDI).

### 2.1.4. Personal Data

Personal data relates directly to physical persons, or allows concluding on different aspects relating to physical persons. In addition to general personal data such as address and age, further examples are given by bank details, hair color or dress size. Personal data is subject to the General Data Protection Regulation (GDPR) and may potentially be generated in companies, offices or even in research.

In case of a personal reference within a dataset, the further usage is generally limited due to privacy protection issues. The data protection regulations legitimize extensive processing only with the existence of a legitimate legal basis and compliance with the data protection principles required by the GDPR. Special care should be taken of so-called "special personal data" referring to particularly sensitive data such as information on ethnic and cultural origin, political, religious and philosophical beliefs, health, sexual orientation and further.

In the case of personal data, physical persons are entitled to "informational self-determination". For companies, authorities or third parties in general, the storage and processing of personal data is therefore only permitted with the consent of the data subject, i.e., the person the data refers to. In addition, physical persons have the right to inspect the data stored about them within

companies/authorities and to initiate its deletion if necessary. Companies and authorities that want to process personal data, or even use it commercially, must pay special attention to belonging data protection issues and regulations.

### 2.1.5. Behavioral Data

Behavior data is given by digital data of/from citizens, which emerges based on their behavioral patterns. Such digital information is based on automatically generated/obtained samples based on the behavior of citizens involved with some sort of sensor equipment (e.g., heartbeat sensors, GPS, smartphone-based sensors, etc.). Data obtained in a machine-generated or automated manner remains property of the corresponding citizens, no matter if it is anonymized or personalized, or if the data has been further processed and new information has been produced through interconnections and data center computations.

Recent German data eco-system studies [10] handle the "behavior-generated personal data" from the perspective of individual property. Behaviorally generated personal data must therefore be clearly distinguished from the data protection law term "personal data". In other words: the main subject of data protection legislation is personal data, whilst the subject of data ownership legislation should be correspondingly given (in this context) by the behavior-generated data. Existing legal loopholes on the subject of "data ownership" call for the introduction of a primary intellectual property right/law for behavior-generated information of citizens. German consumer organizations are therefore calling for a clear classification of data, in order to determine the scope of ownership, and correspondingly the scope of the exploitation rights [10].

### 2.1.6. Freely Available Data

The term and the need for "freely available data" is closely related to the "open data" and "open government" movement (as illustrated in Figure 1), which build on "freely available data" as follows: (a) in general, significant impulses for the improvement of political, social and economic data promoting social cooperation are expected (keywords: participation, transparency and cooperation) [11]. The open data/government movement continues to assume that freely available data (b) contributes to better forms of governance (i.e., better governance in general) and (c) provides various added values for policy, administration and citizens at the procedural level, for example by promoting "open innovation". Freely available databases have great innovative potential for business, administration and society, as well as for social innovation and economic development [11].

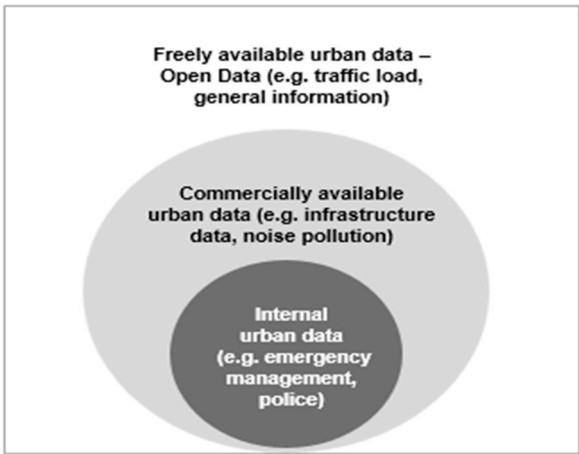

**Figure 1.** Data Layers in an Urban Data Space.

It should be noted again that the term "freely available data" is not synonymous with the term "open data". "Open data" is when public data is freely available to the public without restrictions.

Data provisioning and usage subject to restrictive licensing terms, that is, usage restrictions opposing established open data licenses, contradict the understanding of open data [12]. In such cases, we have to deal with "closed data" or "shared data".

As part of the smart city activities of recent years, "open data" has been given high priority. Various "Smart City/Community" initiatives [13–16] have been advocating for urban development in the direction of urban digitalization and ICT-based ecosystems for urban environments; both in Germany and in the broader European context. This is expected to be realized with the help of administrative data and its provision or "opening" via corresponding open data portals. Examples are the open data portals of Berlin (daten.berlin.de), Cologne (open data—koeln.de), the transparency portal Hamburg (transparenz.hamburg.de/open-data/) and GovData.DE as the data portal for Germany, which was initiated by the Federal Ministry of the Interior. Since such data are usually already available in the municipalities, many of the "smart city/community" pilot projects carried out throughout Europe in recent years have called for the completely free opening of public databases. Many ICT research projects initially had to limit themselves to open data as key data input, or to generate data by crowd-sourcing for the concrete project needs, since other data was not available for technical, licensing or business reasons. For example, EU projects in the area of Sustainable Mobility showed interest in the mobility data coming from navigation system manufacturers [17]. This data could not be used in the projects in general, because it was not freely available for research activities.

The open data concept continues to play an important role in the context of the establishing urban ecosystems and platforms as well as in all major international and national strategies, roadmaps, collaborations and standardization relating to this topic. Further basic definitions of open data are given by the preliminary study on the GovData.DE portal and by the Sunlight Foundation [18]. The open data activities in Germany are promoted by studies such as the current report [19] of the "Technology Foundation Berlin" on the status quo of open data in the Berlin administration. Among other things, this study notes that there is often a lack of clear responsibilities for the subject of data publication and provisioning. In this sense, it is recommended to set up a special body to coordinate the data publishing/provisioning activities of the municipality. This body would need to have the necessary authority to view the databases and prepare them for reuse within the individual offices and units; such processes are of high importance for systematic data publishing and provisioning.

### 2.1.7. Commercially Available Data

Commercially available data—as pictured in Figure 1—can be generated by either private or public agencies/institutions. Typically, this type of data is provided by companies that are interested in selling the data. Accordingly, the "London Data Strategy" [6] defines "commercial data" as being distributed under a license that allows re-use and processing strictly only in exchange for a monetary payment. Companies that sell commercial data are (for example) map or navigation system manufacturers, energy companies, mobile service providers or even post companies. For example, mobile communication providers can sell analysis data of traffic and mobility streams based on anonymous wireless network signaling data. This data enables so-called "geo-marketing insights", i.e., insights into urban matters that previously could not be analyzed, or could be understood only with great effort. These findings can be of significant interest to different actors (industry, service providers, political parties etc.). For instance, such data can provide information about the number of road users traveling between two districts or cities. Based on such information, the volume of traffic on a route in the ecological and social sense can be influenced. In particular, similar evaluations can significantly support strategic decisions and operational improvements in transport and other sectors.

In general, companies do not distribute primary data (for example, customers' activities within the mobile network). Normally, commercially available data is published as secondary data. Secondary data is generated after further processing steps from primary data. Possible processing methods of primary data can be: aggregation, generalization, interpretation and classification. For example, (primary) data is often tailored to specific customer needs, and anonymized and filtered based on data

protection legislation, so that no conclusions are drawn to individual persons and thus the privacy of the citizens can remain protected. Data driven enterprises or authorities spend considerable resources (for example, for data analysts) and infrastructure (such as high-performance computing capacities), which have to be financially reflected in the final price for data.

Currently, the question remains as to how so-called behavior-generated data, which is often commercial data, can ideally be provided as part of an Urban Data Space. So far, the problem in the area of corporate data is solved—e.g., in the context of traffic planning such data is handled based on jointly concluded contracts between data companies and the city/municipality. Currently, the question of data ownership for behavior-generated data of citizens is discussed in a German debate on the introduction of a new "data ownership" legislation [10].

### 2.1.8. Internally Available Data and "Publicly Unavailable Data"

Internal data are generally those data which are available within the authorities, companies or reside in private ownership, and which for various reasons cannot or should not be made available to the public as raw data—see Figure 1 for the position of such data within the data layers of an urban environment. Mostly, it is data that is intended for internal organizational purposes. Hence, publication and external use is not considered.

In principle, institutional (public) administration data in Germany should be openly available. This is the situation according to the belonging law [20] governing access to information of the Federal Government (Freedom of Information Act—IFG)[3]. It also applies to other federal bodies and institutions as long as they carry out public-law administrative tasks. The belonging authority can and should (upon request) provide information, grant access to a requested file or provide information in any other way.

### 2.1.9. Summary

The current section provided a classification of urban data which is paramount for setting the context and defining the Urban Data Space. In general, a large variety of classification schemes are possible depending on the considered data characteristics. The classification provided here encompassed the *official institutional data* (i.e., public sector data) and moved over to describing the properties and the data sharing conditions of *enterprise data*, which is of key interest in an urban environment. Further, relevant data is classified in terms of its origin as *research*, *personal* and *behavioral* data, while at the same time it can be seen as *freely available*, *commercially available* and *internally available* in regard to its access characteristics. Thereby, in the scope of the personal data classification, special emphasis was set on recently established regulations with respect to data privacy and data protection (i.e., GDPR).

The classification provided in this section is of paramount importance for understanding key dimensions of the proposed Urban Data Space and understanding important aspects and design decisions in this context.

### 2.2. Data Access and Data Sovereignty

The current legal situation (related to the legal rights of data use) is still unsatisfactory [21]. For different types of data, different usage rights apply depending on the context and varying between the domains. For years, a national and international political discourse has been ongoing about these imperfections of data laws and the potential creation of new legal frameworks for data ownership [22]. As indicated, it is still a matter of debate whether data can even be conceived as individual private property, i.e., whether data can be treated more or less like material or intellectual

---

[3]    In German: "Gesetz zur Regelung des Zugangs zu Informationen des Bundes (Informationsfreiheitsgesetz—IFG)".

property. Furthermore, within the previous paragraphs we outlined the pending loopholes regarding the classification of behaviorally generated data and its ownership assignment. Further issues are related to data exclusivity rights, or to whether clarifying rights to data utilization over contracts is sufficient. It is also unclear in most cases how data access rights can be defined and how this affects the business models and the evolution of a data economy. As part of building the data economy, the complex "data ownership, data sovereignty, data exploitation and data protection" aspects reveal numerous clarification needs.

There have been various interpretations of the term "data sovereignty" in use by political parties in Germany. Basically, the debate about this term deals with the right to handle data relating to a person in situations, which are not already covered by existing regulations. If one explains the term in the light of the various arguments in the social and political discussion, two interpretations best cover the general understanding of "data sovereignty": a) sovereignty in terms of data protection and b) sovereignty in the sense of a property-like view.

### 2.3. The Urban Data Space

The emergence of the term "data space" is recent and related to the emergence of the concept of the "European data economy". In April 2018, the European Commission presented the follow-up strategy paper on the European Data Economy—"Building a European Data Economy" [23]. It places the notion of a data space in the context of a data economy and emphasizes the fact that the digital transformation is not limited to a social scope (such as focusing on economic aspects, for example), but that it encompasses all areas of life. In its communication note, the EU defines the "European Data Space" as "*a seamless digital territory on a scale that enables the development of new data-based products and services*" [24].

Data spaces contain data and serve as enablers for digital services, for example, linked data semantic and web technology-based platforms and services [25]. The term Data Space is applicable only to digital data. Digital data refers to basic data (raw data), value-added data (processed data), meta-data (data describing basic data and value-added data) and derived information (often derived from data by means of combining various datasets towards logically obtaining facts or interpretations); all called data for short. The Data Space applies to this data, but also to technical data stores and various types of data processing. The Data Space can also have a spatial scope. For example, the term "European Data Space" clearly refers to the territory of the European Union.

In institutional and personal terms, one can imagine a Data Space as a network of actors. From a technical point of view, the Data Space is a data infrastructure with technical standards, where data can be securely exchanged and linked between actors in the Data Space. In legal terms, the Data Space can be constructed as a separate entity with rules in a clear legal framework that should be entitled to "data security" as well as "data sovereignty" [9] of its participants. Functionally speaking, the Data Space can be understood as a demand-oriented system that can be actively shaped by its actors [22].

Within the emerging data economy, various decentralized "Data Spaces" can be identified. In particular, Data Spaces differ in terms of spatial, legal and economic objectives. Data Spaces can be identified at European, national, regional or local level, separated from each other in terms of their actors (e.g., industry, municipalities . . . ) or domain-specific (e.g., mobility data space, energy data space, medical data space, research data space . . . ).

As an "Urban Data Space" we refer to such a Data Space containing all kinds of data that may be relevant to the urban community as well as to the urban economic and policy space. Ideally, based on the "smart city/community" concept, it encompasses all data relevant to the municipalities and their stakeholders from all domains (energy, mobility, health etc.) in urban environments that arise in both analog and digital contexts.

The boundaries of an Urban Data Space are not necessarily within a specific municipal space. An Urban Data Space can also be extended to the dimensions of an economic area that is important

for the municipality, as well as to the associated administration, living, but also legal, experience, action, identification, communication and socialization space. The Urban Data Space includes all data generated by persons, companies and/or machines (personal and non-personal) as well as behavioral data (i.e., data generated by human behavior), be it internal, commercial or freely available, provided that it is closely related to the corresponding urban space.

The objectives of an Urban Data Space are: (1) the increased availability and utilization of urban data, (2) improved access to and better transmission of data within the municipal administration, municipal enterprises and other stakeholders, (3) the transparency when handling non-personal data, (4) technically sound concepts for data security/protection and improved data quality, (5) interoperability and standardization of urban databases and communication protocols, (6) the development of municipal and regional data analysis; (7) the promotion of data-based business models in urban areas by the state and municipalities and in promoting development opportunities for innovative business ideas of small and medium-sized enterprises in the municipal area, (8) building a flexible technical IT infrastructure that integrates all available meta-data and data, and (9) the liability and security related to the utilization of innovative technology.

### 2.4. Stakeholders in the Urban Data Space

When building up the data infrastructure for an Urban Data Space, the diverse interests of all stakeholders should be taken into account and their potential involvement and contribution to the overall eco-system supported. The "smart city/community" concept calls for the actors to provide the most interesting possible data (real-time data, big data . . . ) within the Urban Data Space so that diverse and innovative utilization scenarios are facilitated. The "network of actors" of the Urban Data Space can be structured as follows [6]: (1) Structural actors: Urban Data Space operators (UDS operators) are the actors who actively shape the digitalization strategy, operate and promote the data infrastructure and data usage in urban areas (e.g., mayor, public services, private sector representatives, universities, regulators, standardization bodies, ethics council . . . ), (2) Supporting actors: data providers (UDS data providers), providers of IT services as well as public and commercial data providers; organizations involved in the provisioning and operation of the urban infrastructure and handling the data (e.g., municipal companies, telecommunication companies, private sector, transport networks . . . ), (3) Contributing actors as users of the Urban Data Space (UDS users) and the belonging data infrastructure. These could be stakeholders such as developers of data-driven business models or citizens (e.g., data enriching collaborators, integrators, consumers and others). UDS data providers publish data in conjunction with terms of use and charges, or free of charge depending on the underlying operational and business model. The following (data) providers can play a significant role within an Urban Data Space (either based on terms of use and charges, or free of charge): (1) municipalities and municipal companies, associations, (2) commercial enterprises, (3) research, (4) citizens, (5) offices and organizations of the public sector, (6) non-governmental organizations (NGOs) etc. UDS operators ensure the secure and trustworthy operation of the Urban Data Space. UDS users must respect the terms of use and, if necessary, pay fees. For the classification of the urban data, it can be further said that its terms of use may permit only a purely (provider) internal use. That is, use is restricted to a group of providers or users. On the contrary, a public usage can be allowed that is free or requires a monetary value in return.

### 2.5. Smart City and Communities ICT Reference Architecture

In order to optimally structure and sustainably use all urban data in the sense of a smart city, the technical structure of a data platform is required—a data platform can link together all the available urban data. A data platform that extends horizontally across all domains of a municipality and that can access all the required data is seen as a fundamental part of a Smart City/Community ICT infrastructure. An appropriate data platform as a database is included in all ICT reference architecture models for smart cities.

In the past few years there was increased research effort relating to the concept of an ICT reference architecture for "smart cities and communities". Various ICT reference architectures have been developed and tested in a number of European and national research projects. Reference architectures and models are increasingly used in telecommunications and the Internet domain, thereby enabling global networking and communication of data, video and voice. Two of the most prominent reference models are given by ISO/OSI [26] and TCP/IP [27], which have unified telecommunication and Internet communication architectures and provide a common understanding for sustainable development of global communication technology. The development of these two protocol families has facilitated the Internet and the digitalization of our societies in the first place. In particular, ISO/OSI and TCP/IP protocols provide inter-device interoperability—e.g., ranging from switches, routers, media gateways, to end user terminals such as smartphones, tablets, and desktops—from various manufacturers.

Another reference framework that has gained relevance in recent years is TOGAF (The Open Group Architecture Framework) [28], which is increasingly being used in the development of enterprise architectures. TOGAF has inspired some of the key activities/collaborations on urban ICT reference architectures in recent years—for example DIN SPEC OUP 91357 [16] and EIP SCC [15].

With respect to reference architectures for ICT in smart cities, the Smart City Charter of the German Federal Institute for Research on Building, Urban Affairs and Spatial Development (BBSR) [3] explicitly states that the utilization of open interfaces and standards in the urban context will lead to a structured and flexible way of digitalizing a municipality. A reference architecture will promote the integrative cooperation of several vendors and support the sustainable extension/enhancement of the ICT infrastructure through new software and hardware modules.

The European Initiative "The Marketplace of the European Innovation Partnership on Smart Cities and Communities" (EIP SCC) [14] summarizes the concept of a reference architecture as a tool to support the smart cities and communities, an abstract IT-technical perspective for the realization of an urban ICT infrastructure. On the other hand, the abstract approach of the reference architecture as a blueprint allows for the consideration of specific needs of the community by strengthening the resulting real technical architecture of the community/municipality/city—as a result of the reference architecture through standards, open interfaces and interoperability aspects. Moreover, it is a basic assumption that an ICT reference architecture integrates or connects existing ICT solutions in the Urban Data Spaces. Existing systems should remain in place, but at the same time fit into the new structure. This requires improving their interoperability or the interoperability of the entire existing technical architecture within the municipalities. The main goal of a generic reference architecture is to enhance these real technical urban IT architectures and enable their sustainable extensibility and scalability, while at the same time reducing dependence on individual vendor/operators (vendor lock-in effects). Finally, a reference architecture provides a common terminology that applies to all urban spaces and enables technical discussions between different actors.

According to our understanding, we define an ICT reference architecture as follows: An urban ICT reference architecture sets itself the goal (1) of describing an abstract structure of the ICT infrastructure and related interactions, especially between the utilized ICT components. Based on this abstract structure, (2) an ICT reference architecture creates an ecosystem for information and communication technology in the urban context, in which various actors can participate. This ecosystem is (3) open to SMEs, large corporations, open source initiatives and related open source software modules. By applying a reference model, the (4) classification and interaction of different ICT components is supported via open, standardized interfaces/APIs within the reference architecture, such that the (5) implementation of "Smart City/Community" scenarios based on integrative solutions is ultimately enabled. In addition, the continuous (6) extension of the municipal ICT infrastructure is ensured by the addition of further components according to the rules of the applied reference architecture. In particular, by using a reference architecture, it is possible to (7) replicate ICT-based smart city/community solutions between municipalities, and (8) further exploit the combination of

components from existing smart city solutions and adopt in an integrative pattern into new innovative urban services and applications. It is particularly important to note that an ICT reference architecture does not pursue a disruptive approach, but an evolutionary one that (9) takes into account the existing ICT systems and maps/positions them within the framework of the reference model.

The frame for the Urban Data Space provided in this chapter constitutes a solid prerequisite for following the analysis of the situation on selected German cities as well as for defining the Urban Data Space and its Data Governance and Sovereignty aspects in the coming chapters. Thereby, the needs identified in this and the coming chapters are specifically addressed during the design of the abstract blue print architecture of an Urban Data Space with its recommendations and characteristics.

## 3. Analysis of the Situation in selected German Cities

When choosing the model municipalities, various aspects were considered of particular importance: Interested municipalities should have an advanced status with regard to the systematic management of urban data and the existence of established IT departments. It was also important that the municipalities actively shape the interaction between administration, urban society, science and digitalization activities, and express an interest in actively supporting the study. Furthermore, recommendations from the German Organization of Municipal Enterprises (VKU—Verband kommunaler Unternehmen) on interested municipalities were taken into account. Finally, we paid attention at selecting cities and regions of different sizes and types (e.g., port city) as well as different characteristics of integration in the surrounding region, in order to achieve a broader view on various aspects of relevance.

As a basis for carrying out the analysis, an inquiry form was first designed to query various aspects of the Urban Data Space. It serves to record the IT systems that municipalities and municipal companies work with. Based on this, the data—with which the systems work—was analyzed in further detail. In addition, the questionnaire includes sets of questions on strategic aspects of the Urban Data Space, such as collaboration, strategic frameworks (concepts and documents), and the potential use cases for the available data. The legal part of the questionnaire includes legal framework conditions, as well as licenses and data usage rights/rules in place within the involved municipalities. The questionnaire can be found in [5].

The contact persons in the municipalities received the questionnaires with the request to complete them independently, as far as possible. Based on this, (semi-)structured interviews were conducted with various employees from local government and municipal companies. The selection of the persons to be interviewed was carried out by the municipalities.

Based on the results from the conducted interviews and performed analyses, different hypotheses are drafted which contain generalized statements for the data situation in the German municipalities. These statements serve as the basis for recommendations considering the strategic framework, the data diversity, the cooperation, the IT infrastructure, the interoperability and economic aspects of data utilization towards establishing viable Urban Data Spaces. The following paragraphs contain only a high-level summary of the situation in the selected German cities as well as the recommendations on German national level for the cities/municipalities to consider towards the implementation of their Urban Data Spaces. The detailed results of the conducted studies can be found in [5].

### 3.1. Emden

The situation analysis in Emden shows that the city is pursuing a structured methodological approach to digitalization. In particular, based on an established digitalization roadmap, an important tool is put in place, which represents a "manual" for the execution of the local smart city project.

The selected approach regarding the identification of suitable business models is an interesting way to kick-start the activities towards establishing an urban data platform and correspondingly an Urban Data Space. Whilst normally new technologies (e.g., data platforms) are directly introduced in German and European cities, it is first checked and analyzed in Emden whether the introduction of an IoT platform can be supported by sufficiently viable business models. Since not yet existing in

Emden, the introduction of commercial and crowd-sourced data could provide corresponding potential within the design of urban data and belonging platforms, and ultimately in the identification of other business models. The previously selective exchange of Emden with the municipality of Monheim or other municipalities, which already use commercial data, could be extended into a strategic cooperation, so that in the long term the experiences can be re-used and new ideas can be generated together.

### 3.2. Bonn

The performed analysis and interviews show that the city of Bonn is planning and accelerating the topic of digitalization, in order to remain attractive for citizens and the industry in the future, and to increase efficiency in the administration. This can be seen, for instance, in the project "Digital Administration" as well as within the newly created central coordination office/position for digitalization topics—the "Chief Digital Officer (CDO)". Having already pioneered the open data context, the city of Bonn is still interested in taking part in new developments at an early stage and helping to shape them. This becomes apparent, among other things, in the smart city test areas in the city.

### 3.3. Dortmund

Within the context of Dortmund, based on the need to accomplish dynamic and more complex tasks in the context of Smart City/Community, the availability of real-time data in various areas—above all transport, energy, or security/safety—is of paramount importance and should be considered an aspired goal of the smart city considerations in Dortmund.

A possible marketing and sales strategy for data, which are not to be assigned to the context of open data and where a monetary value is expected, has not been yet developed extensively in the city of Dortmund (such development is expected mainly from the municipal companies). Individual fee models for certain types of data and information are available, but a holistic cash-flow model does not exist. Concerning the data, which a city like Dortmund (mainly its municipal companies) could provide against monetary payments, a continuous balancing of the expenditures for the supply and marketing with respect to the expected income is to be considered. Under certain circumstances, other factors are to be taken into account for some data—these factors might influence a decision against a commercial marketing of the datasets (e.g., social aspects and overall benefits of critical value for the population and community as a whole). Despite the above considerations, the involved representatives of Dortmund believe to some extent that no uniform fee or cash flow model will be established across different municipalities and domains.

In general, the setup of an Urban Data Space is seen by the city of Dortmund as a helpful overall construct, which can support the above goals and aspirations in the context of Smart City/Community.

### 3.4. Discussion

Based on the results of the interviews and belonging analysis, the corresponding recommendations for action are listed below and explained briefly. Thereby, the recommendations are based on gaps identified during the (semi-)structured interviews with key persons in the municipality in question, whereby a set of hypotheses were defined in advance and validated against the obtained interview results (using statistical methods where appropriate). Specific details on the process go beyond the scope of this paper and can be looked up in [5].

The following constitutes the condensed list of recommendations based on the study and belonging procedures in [5]:

- Identification of further strategic fields of action for a comprehensive strategy for an Urban Data Space
- Systematic inventory of municipal data and local ICT infrastructure based on the German DIN OUP 91357 ICT reference model for smart cities

- Development of new data sources and raising awareness that urban data is a valuable resource
- Awareness of the presence and potential of crowd-sourced and crowd-sensed data, among others for the urban operations and services
- Raise awareness of the presence and potential of social network data and increase the systematic exploitation of social network activity as a strategic resource
- Involvement of all relevant actors and stakeholders in the construction of an Urban Data Space
- Introduction of a data officer for the Urban Data Space as a dedicated position/office
- Strengthening or establishing a higher-level coordinating body for digitalization
- Introduction of a common terminology of the Urban Data Space to facilitate cooperation between actors
- Structuring and strategic development of municipal/urban ICT infrastructures
- Transfer of the existing municipal technical infrastructure into a standard-based infrastructure with open interfaces and formats according to a general ICT reference architecture such as DIN SPEC OUP 91357 based on EIP SCC
- Consideration and integration of the specific local needs and requirements of a municipality in the construction of an urban data platform
- Awareness of possible dependency issues (vendor lock-in) and early actions to avoid such potential problems
- Creation and formulation of an offer that supports and accompanies the installation, data provisioning, usage and operation of an Urban Data Space
- Usage of existing and provisioning of own open source software components in an integrative manner towards the realization of an Urban Data Space
- Analysis of a large number of potential urban use case based on data utilization

In order to address the above recommendations, a general structure of an Urban Data Space is required. Such a structure is provided in the next chapter of this paper, which deals with the design of an Urban Data Space. Thereby, this structure is based on the framework of concepts relating to an Urban Data Space (provided beforehand) in combination with the identified recommendations in this chapter.

## 4. Designing the Urban Data Space

The technical prerequisite for a functional Urban Data Space is the coherent, coordinated and networked data and system landscape of all actors, departments and organizations of the involved municipality.

The analysis presented in the previous chapter shows that the data and system landscape in the municipalities is very fragmented, as well as is the associated existing knowledge. It must be emphasized that the technologies and technical concepts are already existing, both for overcoming the fragmentation of the data aspects as well as for the hardware and software integration or orchestration for the purpose of creating a common Urban Data Space. The present legal framework also offers development opportunities for municipal business models.

However, municipalities have so far lacked a holistic concept for the permanent and sustainable construction of Urban Data Spaces. This chapter presents a practicable technical approach and advocates for the application of a "standardized, open reference architecture" as a blueprint for the construction of Urban Data Spaces in German municipalities.

An open reference architecture—as described for example in DIN SPEC OUP 91357—is characterized by its integrative and modular character. It fulfils principles such as interoperability, reusability, openness and scalability. These design principles for IT architectures in public administration have been identified and promoted by SAGA [29] as a key "eGovernment standard" (since 2002) by the Federal Government's Information and Communication Office in the German Federal Administration (Bundesverwaltung). These principles are also used in open reference architectures for smart cities/communities, as in DIN

SPEC OUP 91357 [16]. The application of SAGA also ensures that the selection of technologies is based on transparent criteria and consistent quality requirements. In addition, we suggest that the ICT components of an Urban Data Space used in specific implementations of DIN SPEC OUP 91357 should be audited and certified as required by BSI's [30] security requirements. The compliance to the BSI[4] security requirements and the design principles of SAGA—in the context of DIN SPEC OUP 91357—ensure the security, resilience and trustworthiness of the Urban Data Space.

The key benefits of this approach to building urban datasets are: (1) the systematic structuring of existing ICT solutions and datasets along the blueprint image, (2) identifying the gaps in the city's ICT architecture and the needed actions. At the same time, it becomes visible which systems or components already exist and how they can be linked and mapped to the blueprint reference architecture. (3) The openness of the interfaces and formats promotes interoperability as well as the reuse of components and solutions. Existing legacy systems can be integrated by providing and interfacing with interoperable interfaces. (4) Existing ICT components from other communities can be interchanged and reused. (5) A standards-based approach with open interfaces and formats promises enduring, future-proof, high-security ICT solutions. (6) On the basis of the general reference architecture, it is possible to develop a municipality specific Urban Data Space that fulfils the locally defined requirements for a concrete municipality in the long term.

These benefits of using DIN SPEC OUP 91357 will be discussed in more detail below. The necessary steps for the establishment of an Urban Data Space are discussed according to the method outlined above. Next, the objectives of the technical approach are formulated and a motivation is provided for the use of established ICT reference architectures for Smart Cities. Based on this discussion, the benefits for municipalities are derived and specific technical artefacts required for the implementation of an Urban Data Spaces are described.

### 4.1. Required Steps Towards the Establishment of an Urban Data Space

The interviews with actors in the municipalities involved in the studies give a picture of a fragmented technology landscape. This applies to the heterogeneous datasets in the inventory and their availability in the context of an Urban Data Space. In particular, there is a lack of conception and systematic structuring of the existing ICT landscape. Furthermore, the potential difficulties in establishing Urban Data Spaces are not approached in a systematic way, taking into account existing components and ICT systems in the investigated municipalities. Possible steps to deal with the current situation and to enable further development towards an Urban Data Space are listed below. These steps will later be presented separately as recommendations for action and aim at the sustainability of the proposed concepts.

In order to develop the Urban Data Space sustainably and incrementally, one must begin with the systematic review and inventory of the data space and the mapping of the locally existing technical structure as exemplified in Figure 2. In particular, it is required to gain a complete picture of the ICT systems in operation, historical databases, as well as the currently available, generated and consumed data through local community services and applications. In addition, the associated interfaces and data formats of all systems must be described and classified accordingly.

---

4    BSI stands for Bundesamt für Sicherheit in der Informationstechnik/Federal Office for Information Security.

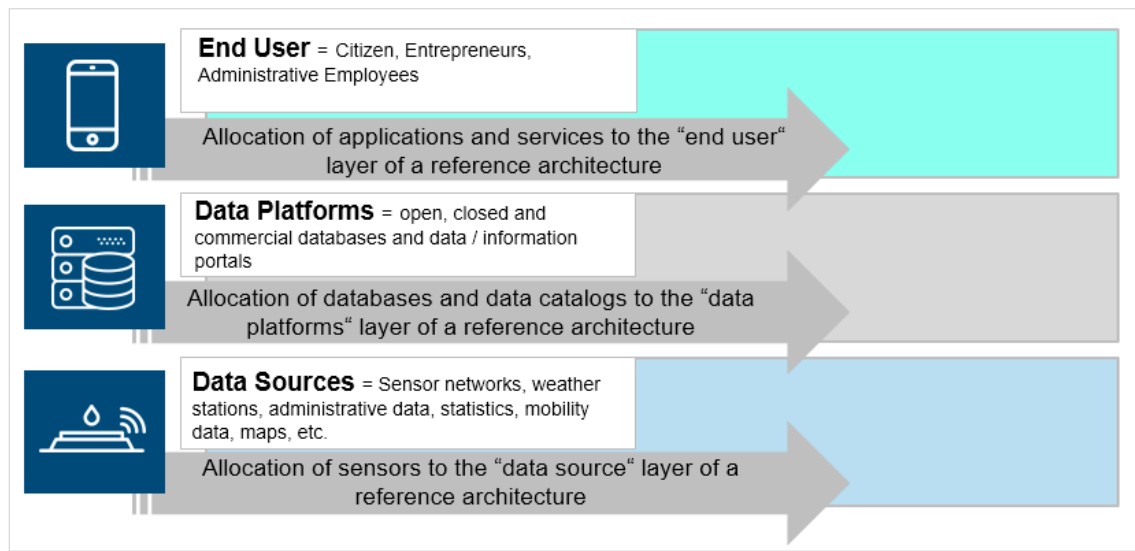

**Figure 2.** Structuring an existing ICT landscape of a municipality based on an ICT reference architecture.

As a result, the aim is to classify the local technical inventory into a general architecture. Our recommendation is to use ICT reference architectures designed especially for smart cities/communities—such as EIP SCC [15], DIN OUP 91357 [16] or the "Triangulum" [31–34] and "Espresso" [32,35] reference architectures from the belonging European Horizon 2020 research projects [36]. The use of such reference architectures enables the systematic development of a municipal data and system inventory in the direction of an advanced Urban Data Space, taking into account and integrating the specific needs and requirements of a city/municipality in the construction of an urban data platform. In addition, openness—in terms of open interfaces, open data, data models, and open standards—as the basic principle of an ICT architecture, enables the involvement of all relevant actors in the design and construction of the Urban Data Space. The open concept of a reference architecture composed of many interchangeable modules enables a vibrant and dynamic ecosystem of coexistence across multiple products and companies. The open urban platform is free of so-called "vendor lock-in effects"—which means that its modularity and the interoperable interfaces between the modules greatly reduce and at best prevent dependency on individual manufacturers and operators. The concept enables the participation of a large number and variety of involved actors: the local IT, small and medium-sized enterprises as well as start-ups, large-scale industry, the open-source economy, various initiatives as well as citizens. The concept of an open urban platform enables many initiatives and companies (including SMEs) to set up pilot projects in cooperation with municipalities or economically-linked municipalities, thereby promoting the sustainable development of an Urban Data Space and smart urban scenarios on top.

*4.2. Technical Goals*

Along the steps for the establishment of an Urban Data Space, corresponding goals for the technical implementation can be derived. In the first place, a technical implementation of an Urban Data Space based on a standardized and expandable ICT reference architecture for smart cities/communities is to be developed. This should be based on open interfaces and open formats. It is also recommended that local authorities expand their available amount of open data and promote the utilization of open source components. This will create an ICT ecosystem for Urban Data Spaces that will allow cities and communities to avoid widespread problems such as vendor lock-in, i.e., dependency on large platform manufacturers, and to ensure competition and data protection.

Existing municipal ICT systems can serve as a basis for creating an Urban Data Space. Existing implementations and artefacts should be captured as components and integrated into the overall

technical infrastructure of the Urban Data Space. Missing components should be systematically added to maximize data and information exchange within the urban data platform and provide as many innovative services and offers as possible. In this case, the used ICT components are to be regarded as part of a security-relevant infrastructure and to be evaluated for productive operation in accordance with the BSI cyber-security requirements as well as to be examined for potential vulnerabilities.

The use of reference architectures for municipal ICT infrastructures is already happening on a broad scale. Many initiatives for smart cities/communities develop solutions based on specific reference models. Especially at the European level, various projects and collaboration initiatives (such as Espresso, Triangulum or STREETLIFE) have developed municipal solutions based on reference architectures. In addition, the European Innovation Partnership on Smart Cities and Communities (EIP SCC) at the European level and DIN SPEC OUP 91357 in Germany constitute two important initiatives that define a so-called "open urban platform". We recommend the construction of an Urban Data Space based on ICT concepts in the sense of a reference architecture as elaborated in EIP SCC and DIN SPEC OUP 91357.

*4.3. DIN SPEC* 91357 *"Reference Architecture Model Open Urban Platform"*

In the following, we will go into more detail about the DIN SPEC OUP. The DIN specification (SPEC 91357) "Reference Architecture Model Open Urban Platform" is the version of the EIP SCC reference architecture adapted for Germany. The rough structure of the DIN SPEC OUP and according to the EIP SCC ICT reference architecture is shown in Figure 3. Since both activities operate on an international level, DIN SPEC OUP has the potential to be considered within the framework of ISO for international standardization.

The DIN OUP ICT reference architecture is divided into eight layers and two columns. Each of these layers/columns has a number of capabilities that are to be realized as part of the layer/column. Detailed lists of the performance characteristics of each layer can be found in the corresponding DIN specification [16] and the European documents on the EIP SCC reference architecture [15]. The lowest layer (0. Field Equipment/Device Capabilities) contains most of the data sources within a community. In particular, various sensors and measuring stations are located there, which generate data for the upper layers of the reference architecture. This is the next layer (1. Communications, Network & Transport Capabilities), which includes the networking of individual devices over a communication infrastructure and stands for the communication network (telecom network or Internet) and the transfer of data from the lower layer to the data platforms in the upper layers.

The 0-layer devices and the first layer communication infrastructure are controlled by protocols and software modules included in the second layer called Device Asset Management & Operational Services Capabilities.

Based on this basic infrastructure, the data sources are networked with the data platforms in the third layer (3. Data Management & Analytics Capabilities). This layer includes data management systems, databases, open data portals, and cloud platforms that store or properly describe the data from the sources (e.g., meta-data catalogs such as CKAN) and provide the data to other services and applications in the urban and municipal context. The data is stored according to its validity (for example, temporary sensor data from the Internet of Things) or versioned and archived according to pre-specified rules. Additionally, in this layer, the data is analyzed and correlated. Moreover, the data can also be provided on the basis of further processing, for example on the basis of statistical algorithms or on the basis of processing in the sense of machine learning.

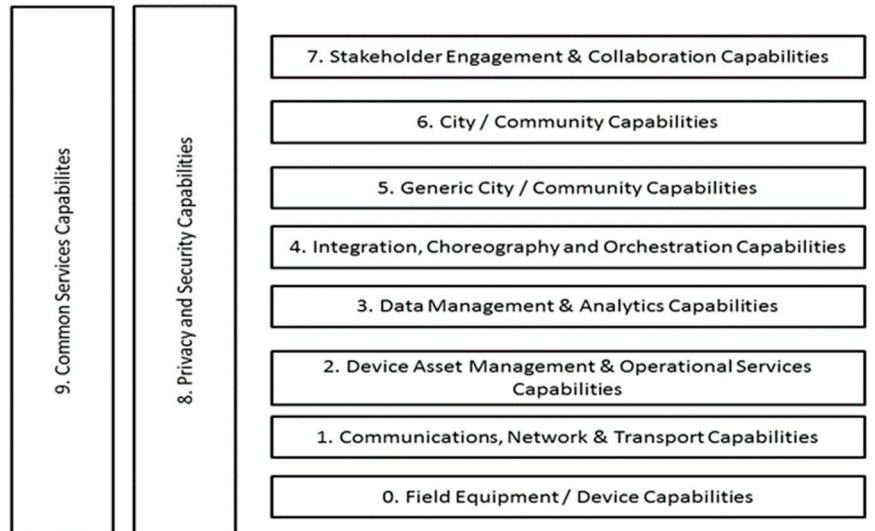

**Figure 3.** An ICT reference architecture for Smart Cities based on EIP SCC and DIN OUP.

The layer (4. Integration, Choreography and Orchestration Capabilities) contains various types of services that offer innovative use cases within a community through the interplay and use of different data and information from the underlying layers.

The following layers: 5. Generic City/Community Capabilities and 6. City/Community Capabilities stand for the various urban processes, everyday activities and general innovations that are made possible on the basis of DIN OUP's ICT processes. Examples include the potential for improving administrative processes and optimizing public transport routes for improved mobility within a municipality.

The seventh layer in Figure 3 deals with the interactions—in the technical, social and economic sense—with the users of the application scenarios and the associated integrative solutions. This is the layer in which the benefits of an Urban Data Space based on DIN OUP become real. In the corresponding applications (such as smartphone apps, information portals, issue management systems, collaboration systems etc.) added value is created for the administration and also for the citizens of a municipality

The two pillars on the left side of the reference model are responsible for privacy and security (8. Privacy and Security Capabilities) and overall system management (9. Common Services Capabilities) for the emerging comprehensive integrative ICT solutions within an Urban Data Space. They cover several layers and ensure IT security and the proper operation of the Urban Data Space.

Finally, it should be noted that the exchange of data and information between the components of the Urban Data Space should be determined by the use of standardized communication protocols and data models. This requirement is explicitly emphasized at European and German level and is the basic prerequisite for the implementation of an open, inclusive, extensible and structured Urban Data Space.

### 4.4. Advantages for Cities and Municipalities

The use of ICT reference architectures described above has many advantages for municipalities: (1) The openness of the architecture—as well as the use of open standards, interfaces, formats and data models—encourages the municipality to reduce dependency on individual manufacturers and operators and thus reduce the risk of vendor lock-in. Vendor lock-in is generally understood as the full dependency of a customer on a particular ICT provider, manufacturer or infrastructure manager. A vendor lock-in arises when, for example, service providers build on proprietary (that is, on non-open and not freely available) interfaces and data formats and thus sell a closed solution as a complete package, for example to municipalities. Such a complete solution can extend over several layers of the presented reference architectures. In such a case, the maintenance—i.e., the bug fixes, but also the updates (software and hardware updates)—is completely in the hands of the corresponding provider and cause permanent costs due to a dependency of the community on the provider with no chance

of improving the situation. Such a situation violates the "open" platform thinking of an Urban Data Space and may make it difficult or even deny regional SMEs access to a community's ICT ecosystem. This situation is also detrimental to a municipality's claim to sustainability. Closed commercial platform solutions can also jeopardize the sovereignty of communities over their data and restrict or prevent free access to urban data. In the event of a vendor lock-in, some data may become the property of the relevant platform operator and may only be accessible at a corresponding cost. This would give municipalities limited opportunities to participate in the use and refinement of their own data.

It should be stated that the realization of Urban Data Spaces using standardized reference architectures with open interfaces and formats—in particular DIN OUP 91357—builds up a (2) support of local self-government and the sovereignty of a municipality over its data—this point can be seen as a significant advantage of the presented technical approach. Based on this consideration, there are further obvious advantages for the municipalities: (3) Local SMEs can implement specific municipal requirements—the openness of the Urban Data Space makes it possible for local SMEs to be assigned specific tasks and developments at any time. (4) Easy integration of digital participation forms and initiatives—this point arises from the openness and the systematic expandability of the Urban Data Space. In particular, the use of open source solutions as well as the consideration of specific needs of the society is possible. (5) The integrative approach supports a consistent cyber-security concept—the openness of the user interfaces makes it easy for different actors to perform certain types of tests that assess and improve the security of the utilized components.

The openness of the architecture enables institutions such as the Federal Office for Information Security (BSI) and related certification bodies to assess the security of an Urban Data Space and to demand corresponding changes that increase cyber-security. In addition, it is possible to use standardized "test suites" that verify and check compliance and security—functional security as well as defense capabilities examined by penetration tests—of the IT systems to be deployed.

From the discussions so far, there is automatically a (6) location (in terms of specific city/municipality) advantage through an improved ICT infrastructure as well as improved (7) interoperability and the possibility of using open interfaces. In addition, an Urban Data Space based on DIN OUP 91357 promotes the use of (8) standardized components based on reference architectures.

Other benefits derived from the envisaged Urban Data Space are already formulated indirectly—notably through the avoidance of vendor lock-in, the involvement of local SMEs, the use of open standards, and improved interoperability and security. All these aspects help a municipality in (9) promoting its general sustainability and (10) preserving its possibilities to act in an independent way when it comes to data related topics.

These benefits are important pillars for the digitalization and development of local communities, and for the exploitation of the data resources that can emerge in an urban environment and can contribute to improving the quality of life and work of citizens.

In the following chapter, another extremely important aspect is emphasized, namely the data governance procedure for an Urban Data Space that supports in achieving data sovereignty for key stakeholders within the emerging UDS concept. The data governance and data sovereignty is designed on top of the proposed UDS and is described in one of many possible concrete varieties that have the potential to guarantee a reasonable operation with respect to the data providers' and stakeholders' interests in an urban environment.

## 5. Data Governance & Data Sovereignty

The terms urban data governance and sovereignty are strongly related to identifying the actors involved in the management, provisioning [23] and utilization of urban data, as well as the required communication and interactions among these actors/stakeholders. On one hand, it is required to develop guidelines determining the responsible stakeholders for the belonging processes of data provisioning and management, while on the other hand the variety of existing regulations and rules, which define the way urban data is to be handled, should be considered. Hence, the required processes

for data handling should be implemented by the identified stakeholders (within an organization) and monitored through appropriate structures in order to ensure compliance to belonging regulations. Such structures would also enable the data owners to keep and evolve their sovereignty over the provided data.

An aspect of paramount importance for urban data governance is constituted by the need for a sustainable and adequate organization for controlling data originating from urban environments, thereby paying special attention to the needs of the community, the public administration and the municipal companies. This implies that organizational setups, guidelines and processes must be correspondingly derived and combined as to interplay successfully. Public institutions collect vast amounts of data, which is a valuable resource for the development of innovative digital services/applications, urban optimization and improved policy-making processes.

The EU has already specified a number of legislative measures to open up public databases across the European Union as an important source of information for the data economy. Directive 2003/98/EC on the re-use of public sector information has created an EU-wide framework that facilitates the cross-border provisioning and utilization of publicly funded data and constitutes a viable asset for the development of pan-European data-based products.

For data governance in general, it should be clearly defined which roles are relevant for the provisioning and processing of data, and how these roles are to be embedded in the decision-making process. The decision-making processes address aspects like data quality management, data access management, general data management and lifecycle management [37]. In addition, there is the key task of managing the meta-data for the datasets in an urban environment, which is also important in the context of data sovereignty and quality. Considering the above statements, different approaches can be derived for realizing data governance in Urban Data Spaces. In general, very often the so-called RACI notation is utilized to implement governance structures. RACI is an abbreviation for responsible, accountable, consulted, informed [38]. Correspondingly, when defining the roles for urban data governance in a particular context, the definitions should be worked out in terms of the four RACI characteristics for the decision-making area in question. Thereby, it is important to emphasize that these roles differ from those defined in the UDS stakeholder analysis in Section 2.4. The UDS stakeholder roles (Section 2.4) focus on systematically classifying the players in an UDS, whilst the roles defined here outline the different tasks to be executed towards guaranteeing data governance and data sovereignty for the involved UDS stakeholders.

In the above described context, the following five roles/bodies can be defined (based on recent EU level research activities [25]), which are described here based on high level considerations: (1) Data Committee—The Data Committee is a decision body with the key role to define and coordinate directives and decisions. If problems arise within an Urban Data Space, the Data Committee is the body which is expected to work out solutions and track their implementation. (2) Governance Officer—The Governance Officer is part of the Data Committee and disseminates, promotes and monitors the policies and decisions within the organization. He acts as the central coordinator for a specific Urban Data Space within the organization in question. (3) Data Owner—The Data Owner is essentially in charge of one or more datasets from a business perspective. The responsibility relates to various topics such as the framework and regulations for further data set usage or to the quality of the data. In addition, the Data Owner takes care of the legal requirements and is responsible for the aspects of licensing and commercial cost/price of the data. (4) Data Steward—The Data Steward bears the responsibility for implementing the requirements of the Data Owner, e.g., proper (meta-)data management. In general, the Data Steward represents the link to the technical users of a dataset. (5) Technology Steward—The Technology Steward manages the technology platform in place for all the data of a stakeholder. The Technology Steward has to guarantee that the selected technological stack is sufficient to fulfil the data quality requirements. In addition, the required technical support includes aspects of data backup, data security and the (meta-)data archiving.

Regarding open data, the PSI (Directive 2003/98/EC) directive clearly states that data originating from public institutions should be freely published and made available to the society as open data. Thereby, Open Data Platforms support this requirement by offering the means for publishing the datasets and handling the belonging meta-data. Different roles are specified within each data portal/platform with data user, data provider and operator being some of the common roles within today's open data portals. Beyond governmental institutions, municipal companies generate interesting types and amounts of data that can enable useful urban applications and services. However, municipal companies are mostly not solely publicly owned but also hold private shares and aim at achieving business goals which might also be based on selling data. Hence, in these cases only particular datasets would be made publicly available in different forms, while others would be offered in exchange for appropriate assets (e.g., money or services). Therefore, it is of paramount importance to emphasize the importance of standards when publishing datasets as open data. For instance, some initiatives [39] require datasets to have the following properties, which are also listed as key principles within the open data community: complete, primary, timely, accessible, machine processable, non-discriminatory, non-proprietary, and license-free—such characteristics can be to some extent achieved by using proper technical standards such as Linked Data, RDF, CSV, DCAT and XML. Furthermore, the data requires many additional interactions and discussions with those responsible for the datasets within the publishing institutions.

## 6. Conclusions and Future Work

This paper presented the results of a recent study that was conducted with a number of German municipalities/cities. Thereby, the need was identified to setup and create so-called Urban Data Spaces within cities and municipalities in order to reveal the vast potential offered by urban data. Building on the recommendations emerging from the study, the authors classify the various types of urban data and elaborate on the characteristics of the identified data classes thereby relating to legal and monetary aspects.

The analysis of urban data and its belonging classification result in a structured presentation and introduction to the large variety of data within an urban environment. According to its origin, data is classified as *official institutional data*, *enterprise data*, *research data*, *personal data* and *behavioral data*. Additionally, based on its access regulations, data is structured as *freely available*, *commercially available* and *internally available*. Besides the classification of the smart city data, further critical concepts are analyzed and introduced, including the notion of data sovereignty/governance, the relevant stakeholders in an Urban Data Space (resulting in a couple of well-defined roles), and the concept of an urban ICT reference architecture, which is at the heart of the emerging UDS.

Another input of key importance for the abstract specification of an Urban Data Space, is constituted by the investigations which were undertaken by the project team within three selected German communities (Emden, Bonn and Dortmund). Thereby, structured interviews were conducted that led to a number of insights and a long list of recommendations, which were subsequently utilized as requirements for the emerging Urban Data Space concept. The most important recommendations/requirements are given by the need to adopt ICT reference architectures for smart cities based on open standards and interfaces, by the requirement to work out an inventory of urban ICT systems in a municipality based on abovementioned ICT reference architectures, and to define an extension strategy for the municipal ICT and data delivery infrastructure based on reference architectural principles leading to the avoidance of vendor lock-in.

After establishing a definition of an Urban Data Space, the concept is analyzed in detail and a proposition for setting up an Urban Data Space is worked out. The authors propose to setup Urban Data Spaces based on emerging standards from the area of ICT reference architectures for Smart Cities, such as DIN SPEC 91357 "Open Urban Platform" and EIP SCC. Thereby, the paper presents the transformation steps required from municipal perspective (especially in the German context) to successfully implement an Urban Data Space. Furthermore, the paper elaborates on vital aspects

such as data governance and data sovereignty and shows how these would be realized within the proposed approach. This results in the proposition of a set of processes, roles and committees that aim at enabling the data providers in an Urban Data Space to determine, steer and understand data publishing implications, regulations and standards.

With respect to future work, we aim at continuing our standardization work at various relevant standardization bodies and relating to different domains and use cases. In addition, a reference implementation of standard open source components for Urban Data Spaces is envisioned, which would allow for quickly setting up an Urban Data Space and evaluating different scenarios and business models. Finally, the idea of quality assurance for ICT components within Urban Data Spaces is of paramount importance and will be pursued on research level.

**Author Contributions:** Investigation, L.B., N.T. and P.L.; Methodology, S.C.; Supervision, I.S.; Writing—original draft, N.T.; Writing—review & editing, S.C., L.B., P.L. and I.S.

**Funding:** European Commission: 646578.

**Acknowledgments:** This work has been partially supported by H2020-Triangulum project (GRANT AGREEMENT No. 646578) and by the German BMBF project "Urban Data Spaces".

**Conflicts of Interest:** The authors declare no conflict of interest.

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
