# Peer review of "Data Governance and Sovereignty in Urban Data Spaces Based on Standardized ICT Reference Architectures"

_data, 2018_

Reviewer 1 Report

The introduction is quite verbose and should cover a little bit more  the motivation and the contribution of the work. It needs proper extensions.  It should state the motivation of the authors to conduct the present work and the way that it could be assistive to specific applications and systems.

Authors should also state the challenges of tracking and analysing these kinds of data. What are the dynamic natures of these kinds of data? Please highlight and explain this aspect.

The related work needs to present in better detail each one of the works addressed describing its aims (research questions specified.

Finally, the paper has a lot of grammar issues and typos. I would benefit from another round of proofreading.

Author Response

>> The introduction is quite verbose and should cover a little bit more  the motivation and the contribution of the work. It needs proper extensions.  

The introduction was extended as to address the reviewers' comment:

"...
On the way to a modern, sustainable and networked city or community, communities must be accompanied and supported. Hence, a set of tools is required that will navigate cities and communities in achieving systematical introduction and adoption of digital technologies, in order to become a data driven smart urban environment. One such tool could be the concept of an Urban Data Space elaborated and defined as a goal of this work in the following.
The goal and contribution of this paper is to define and specify on an abstract level the concept of an Urban Data Space to tackle the imminent challenges to municipalities with respect to the emerging data driven economies across Germany, Europe and beyond. Thereby, we aim at systematically describing the various data types within a smart city/community and at providing an abstract architecture, which can be instantiated in the form of various technical components, interfaces, data formats and processes towards the efficient implementation of a smart city/community. It should be a main feature of the Urban Data Space concept that it is designed in a way as to guarantee data governance and data sovereignty for the involved partners and stakeholders, with special focus on municipalities within this paper. Furthermore, the concept of an Urban Data Space should be able to accelerate the adoption and introduction of smart city technology across Europe and beyond by providing an abstract blue print and facilitating the development, replication and exchangeability of components, services, applications and process according to open standards."  

-> The extensions contains the goal and contribution of the paper
-> The motivation is also stated in the part of "Hence, a set of tools is required ..."

>>It should state the motivation of the authors to conduct the present work and the way that it could be assistive to specific applications and systems.
-> The motivation is explained in the above text
-> The assistive aspects with regard to urban environments is given in the last sentence of the above text

"Furthermore, the concept of an Urban Data Space should be able to accelerate the adoption and introduction of smart city technology across Europe
and beyond by providing an abstract blue print and facilitating the development, replication and exchangeability of components, services, applications and process according to open standards."

>> Authors should also state the challenges of tracking and analysing these kinds of data.

Section "2.1 Data Classification" was extended by a paragraph addressing the challenges:

"The challenges in data acquisition and provisioning in an urban environment emerge on multiple levels. Firstly, the data sources should be secured and instrumented/convinced as to make their data available to a wider range of stakeholders. This phase might include the interaction with municipal departments, municipal companies, local service providers and enterprises, but also the installation of additional/new sensors across the urban area in question and the deployment of services for data handling and analysis. Secondly, the quality of the data should be ensured as to enable a data driven smart urban environment. Data quality includes mainly the correctness, the timeliness and the machine readability of the urban data. Thereby, the data quality might differ in its criteria depending on the type of data as classified further in this section, e.g. sensor data should be delivered within milliseconds or seconds, as to be usable for near real-time applications, whilst municipal public administration data might be adequate if delivered once in a quarter. Furthermore, the quality of the data delivery infrastructure  is of serious concern with respect to conformance to standards, interoperability between components, performance, scalability, stability of the hardware/software implementations, network bandwidth, cyber-security (in terms of confidentiality,
integrity, and availability), data protection and data privacy given its intrinsic nature as a critical infrastructure. Moreover, the utilization of open standardized interfaces is of paramount importance as to enable future business models and avoid vendor lock-in with respect to data provisioning and utilization. In addition, an Urban Data Space should provide marketplace like means for data exchange including data trading as to stimulate the provisioning and utilization of data for future use cases and business models."

>> What are the dynamic natures of these kinds of data? Please highlight and explain this aspect.

The dynamic natures were addressed in the above paragraph extending "2.1 Data Classification".
They were discussed in the scope of ellucidating on "Data Quality". Especially:

"Data quality includes mainly the correctness, the timeliness and the machine readability of the urban data. Thereby, the data quality might
differ in its criteria depending on the type of data as classified further in this section, e.g. sensor data should be delivered within milliseconds or seconds, as to be usable for near real-time applications, whilst municipal public administration data might be adequate if delivered once in a quarter."

>> The related work needs to present in better detail each one of the works addressed describing its aims (research questions specified.
In the course of reviewing the paper, we sharpened the descriptions of the research work accordingly.

>> Finally, the paper has a lot of grammar issues and typos. I would benefit from another round of proofreading.
We did an additional review of the text, as to address the reviewer's comment.

Reviewer 2 Report

This paper contains relevant and timely paper. There are so many interesting elements, that I found it sometimes hard to gain an overview. After reading my impression is that is to develop and implement a Urban Design Space, but I’m still not sure. This should be clarified and using this as a bases the paper needs to be streamlined.  

The abstract is not clear. The abstract should begin with the research driver, instead with the project. Also a clear research questions/objective should be included and at the end 1 or 2 lines of conclusions are expected. You might want to look at ways to describe ‘structured abstracts’.

How the parts are related becomes not easily clear. I would recommend to connect the parts better by adding some explanation. For example after the literature review, explain how this will be used in the remained of the document.

An overview of the research methods is missing. Also it is not mentioned how the research questions are answered using which research methods. I would recommend to present the research methods directly after the research questions.

2.1 contains an interesting classification of data. I recommend to end this section with an overview of the classification to improve readability.

2.4 should contain a stakeholders overview, however, the real stakeholder overview can be found in. 5. Data governance. My recommendation is to integrate 2.4. in 5. Then the difference between literature background as a foundation and the use of stakeholder analysis becomes clear.

It becomes not clear why these cases are selected.

In 3.4 discussion the conclusions are drawn. It is not clear how these conclusions are derived from the interviews (and the literature).   

In “6 conclusions & Further work” there are hardly any conclusions. There are many learnings in the paper that should become part of this section. The main cause of not having the conclusions here seems to be the lack of a clear research objective. The answers to the research questions should be found in this section.

Author Response

>>This paper contains relevant and timely paper. There are so many interesting elements, that I found it sometimes hard to gain an overview.
>>After reading my impression is that is to develop and implement a Urban Design Space, but I’m still not sure.
>> This should be clarified and using this as a bases the paper needs to be streamlined.

The introduction was extended as to address the reviewers' comment:

"...
On the way to a modern, sustainable and networked city or community, communities must be accompanied and supported. Hence, a set of tools is required that will navigate cities and communities in achieving systematical introduction and adoption of digital technologies, in order to become a data driven smart urban environment. One such tool could be the concept of an Urban Data Space elaborated and defined as a goal of this work in the following. 
 The goal and contribution of this paper is to define and specify on an abstract level the concept of an Urban Data Space to tackle the imminent challenges to municipalities with respect to the emerging data driven economies across Germany, Europe and beyond. Thereby, we aim at systematically describing the various data types within a smart city/community and at providing an abstract architecture, which can be instantiated in the form of various technical components, interfaces, data formats and processes towards the efficient implementation of a smart city/community. It should be a main feature of the Urban Data Space concept that it is designed in a way as to guarantee data governance and data sovereignty for the involved partners and stakeholders, with special focus on municipalities within this paper. Furthermore, the concept of an Urban Data Space should be able to accelerate the adoption and introduction of smart city technology across Europe and beyond by providing an abstract blue print and facilitating the development, replication and exchangeability of components, services, applications and process according to open standards."

-> The Urban Data Space was specified as a main goal of the paper
-> The extension contains the clear contributions of the paper and its motivation

>> The abstract is not clear. The abstract should begin with the research driver, instead with the project.

We extended the abstract at the beginning as to name the research driver as requested:

"European cities and communities (and beyond) require a structured overview and a set of tools as to achieve a sustainable
transformation towards smarter cities/municipalities thereby leveraging on the enormous potential of the emerging data driven economy."

>> Also a clear research questions/objective should be included and at the end 1 or 2 lines of conclusions are expected.
>> You might want to look at ways to describe ‘structured abstracts’.

The abstract was extended at the end as to address the the reviewer's comment:
"Finally, we aim at deriving the proposed concepts in a way that they have the potential to be part of the required set
of tools towards the sustainable transformation of German and European cities in the direction of smarter urban environments
thereby utilizing the hidden potential of digitalization and efficient interoperable data exchange."

-> short conclusion of the abstract
-> short definition of the aim of the propsed concepts, which is further ellucidated in the introduction

>> How the parts are related becomes not easily clear. I would recommend to connect the parts better by adding some explanation.
>> For example after the literature review, explain how this will be used in the remained of the document.

The following paragraph connecting the literature review to the rest of the document was put at the end of chapter 2:

"The frame for the Urban Data Space provided in this chapter constitutes a solid prerequisite for following the analysis of the situation on
selected German cities as well as for defining the Urban Data Space and its Data Governance and Sovereignty aspects in the coming chapters.
Thereby, the needs identified in this and the coming chapters are specifically addressed during the design of the abstract blue print architecture
of an Urban Data Space with its recommendations and characteristics."

The following paragraph connecting the analysis of the situation in selected German cities was extended at the end of chapter 3:

"In order to address the above recommendations, a general structure of an Urban Data Space is required. Such a structure is provided in
the next chapter of this paper, which deals with the design of an Urban Data Space. Thereby, this structure is based on the framework of
concepts relating to an Urban Data Space (provided beforehand) in combination with the identified recommendations in this chapter."

The following paragraph was extended at the end of chapter 4 to connect to the following chapter:

"These benefits are important pillars for the digitalization and development of local communities, and for the exploitation of the data resources that can emerge in an urban environment and can contribute to improving the quality of life and work of citizens. In the following chapter another extremely important aspect is emphasized, namely the data governance procedure for an Urban Data Space that supports in achieving data sovereignty for key stakeholders within the emerging UDS concept. The data governance and data sovereignty is designed on top of the proposed UDS and is described in one of many possible concrete varieties that have the potential to guarantee a reasonable operation with respect to the data providers’ and stakeholders’ interests in an urban environment."

>> An overview of the research methods is missing. Also it is not mentioned how the research questions are answered using which research methods.
>> I would recommend to present the research methods directly after the research questions.

The following paragraph was added in the introduction part of the paper in order to emphasize on the approach:

"To summarize our approach: In order to achieve the definition of the Urban Data Space, we proceed with the following steps.
First, an extensive literature review and urban data classification is provided which leads to setting the conceptual framework
for the upcoming design approach. Secondly, we present a summary of the analysis of the situation in 3 German cities. The cities
are selected based on recommendations from key stakeholder organizations in the German municipal IT and data landscape, as well
as on the different size of the cities and level of their digitalization. The analysis was conducted based on structured interviews
with key persons in the belonging municipality leading to the validation of a couple of theses, as well as to the extraction of key
factual statements and concrete recommendations. Based on these recommendations and the conceptual framework, we systematically derive
the concept of an Urban Data Space thereby highlighting its benefits for cities and communities and defining a possible implementation
of data governance and data sovereignty procedures."

>> 2.1 contains an interesting classification of data. I recommend to end this section with an overview of the classification to improve readability.

We added an extra section 2.1.9 which contains a summary of 2.1 with an overview of the classification.

>> 2.4 should contain a stakeholders overview, however, the real stakeholder overview can be found in.
>> 5. Data governance. My recommendation is to integrate 2.4. in 5. Then the difference between literature background as a
>> foundation and the use of stakeholder analysis becomes clear.

The following sentence was added in chapter 5 as to position, section 2.4 to chapter 5:

"Thereby, it is important to emphasize that these roles differ from those defined in the stakeholder analysis in section 2.4.
The stakeholder roles focus on systematically classifying the players in an UDS, whilst the roles defined here outline the different
tasks to be executed towards guaranteeing data governance and data sovereignty for the involved stakeholders."

As stated in the sentence, the stakeholder analysis and the roles in the scope of data governance differ fundamentaly.
Hence, the authors hold that an integration of chapter 5 and section 2.4 would be best given by drawing the relations between
both ellucidations, as conducted within the above clarification.

>> It becomes not clear why these cases are selected.

We added several explanations why we selected the presented cities. In the introduction part, we added the following:

"Secondly, we present a summary of the analysis of the situation in 3 German cities. The cities
are selected based on recommendations from key stakeholder organizations in the German municipal IT and data landscape, as well
as on the different size of the cities and level of their digitalization. The analysis was conducted based on structured interviews
with key persons in the belonging municipality leading to the validation of a couple of theses, as well as to the extraction of key
factual statements and concrete recommendations."

The high level description of the selection arguments is ellucidated further at the beginning of chapter 3.
Thereby, we added the following sentence at the end of the exisiting paragraph:

"Finally, we paid attention at selecting cities and regions of different sizes and types (e.g. port city) as well as different
characteristics of integration in the surrounding region, in order to achieve a broader view on various aspects of relevance."

>> In 3.4 discussion the conclusions are drawn. It is not clear how these conclusions are derived from the interviews (and the literature).   

The start of 3.4 was extended as follows in order to refer to literature and explain the way the recommendations were obtained:

"Based on the results of the interviews and belonging analysis, the corresponding recommendations for action are listed below and explained briefly.
Thereby, the recommendations are based on gaps identified during the structured interviews with key persons in the municipality in question, whereby
a set of hypotheses were defined in advance and validated against the obtained interview results (using statistical methods where appropriate).
Specific details on the process go beyond the scope of this paper and can be looked up in [5]. The following constitutes the condensed list of
recommendations based on the study and belonging procedures in [5]:"

>> In “6 conclusions & Further work” there are hardly any conclusions.
>> There are many learnings in the paper that should become part of this section.
>>The main cause of not having the conclusions here seems to be the lack of a clear research objective.
>>The answers to the research questions should be found in this section.

The chapter has been extended by quite some text pointing out more concretely on the achievements and conclusions of the research.
Thereby, many sentences were embedded as to precise the conclusions and summary of results thereby emphasizing on key aspects.

Round  2

Reviewer 1 Report

Author(s) have addressed all the questions. Best of luck for future research.

Author Response

Thank you :-)

Reviewer 2 Report

The paper has been improved considerably. The overview has become clear and the argumentation flows has become clear. I have some minor suggestions to further strenghten this paper. 

-Include the goal  (that has been added in the introduction) already in the abstract. This improves readability. If possible try to reduce the size of the abstract (focus on the essence).

-The scientific contribution can be founded in the literature much better. I think that there is a clear contribution, but that this is not clear by not including an indepth review of the literature and showing how this work goes beyond this.  

Author Response

>> The paper has been improved considerably. The overview has become clear and the argumentation flows has become clear. I have some minor suggestions to further strenghten this paper.

>> -Include the goal  (that has been added in the introduction) already in the abstract. This improves readability. If possible
>> try to reduce the size of the abstract (focus on the essence).

The specific goal was added as one sentence to the abstract.

>> -The scientific contribution can be founded in the literature much better.
>> I think that there is a clear contribution, but that this is not clear by
>> not including an indepth review of the literature and showing how this work
>> goes beyond this.

We added an extra paragraph showing the progress beyond state-of-the-art in
section 2:

"2.6    Progress beyond State-of-the-Art
Having presented the theoretical frame for the emerging Urban Data Space, including related work and state-of-the-art, the progress brought by the
UDS concept should be clearly formulated. In general the Urban Data Space constitutes an innovation, which combines various cutting edge aspects from multiple areas/domains such as ICT reference architectures, data science, privacy, data protection, legal frameworks for data and smart cities, as well as standardization, operational and business models. The above listed related works are combined in an innovative manner towards the design and specification of an UDS concept, which is meant to provide a unified framework (currently nonexistent) for the further development of cities and communities towards becoming smart and digitalized in a sustainable way."